# Supra-Tuning: Combining Outlier and Low-Rank Adaptation for Sparse and Efficient LLM Fine-Tuning

## Abstract

Large language models (LLMs) have demonstrated remarkable capabilities but remain expensive to fine-tune due to their size. Recent parameter-efficient tuning methods, such as Low-Rank Adaptation (LoRA), reduce the number of trainable parameters while maintaining performance. In this work, we introduce Super, a novel sparse adaptation technique that selects and trains only a small set of influential weights—so-called super weights—identified via outlier metrics such as WANDA. We show that fine-tuning these outlier weights yields strong performance with minimal parameter updates. Building on this idea, we propose Supra, a hybrid method that combines Super with LoRA, merging sparse and low-rank adaptations into a unified tuning strategy. Our experiments on several LLMs and downstream tasks demonstrate that both Super and Supra outperform existing sparse or low-rank methods alone in perplexity and task performance, while reducing computational and memory overhead.

## 1 Introduction

Large Language Models (LLMs) have revolutionized the field of natural language processing, achieving state-of-the-art results in a wide array of tasks including question answering, summarization, code generation, and reasoning. However, their impressive capabilities come with a substantial cost: full fine-tuning of such models requires considerable computational resources, high memory consumption, and significant storage for each downstream task. This inefficiency becomes particularly problematic when deploying models in real-world settings with resource constraints or personalization requirements.

To address these limitations, *parameter-efficient fine-tuning* (PEFT) methods have been proposed. These approaches update only a small fraction of the model's parameters, leaving the majority of weights frozen. Among the most successful PEFT techniques is LoRA (Hu et al., 2021), which injects low-rank trainable matrices into existing layers. LoRA enables tuning with minimal parameter overhead and has been widely adopted for its balance of efficiency and performance.

Beyond low-rank adaptation, recent work has explored the use of *sparse adaptation*, where only a small, selected subset of existing weights are updated. Methods such as SIFT (Song et al., 2023) and RoSA (Nikdan et al., 2024) select salient weights for fine-tuning based on importance scores or gradient signals. NeFT (Xu et al., 2024) takes this further by identifying and training only the most critical neurons. SpIEL (Ansell et al., 2024) introduces scalable sparse tuning with structured expert layers, and SAT (Ma et al., 2024) proposes sparsity-accelerated training with carefully selected updates. $S^2$FT (Yang et al., 2024) combines sparsity with structured decomposition for efficient and generalizable tuning, while GIFT-SW (Zhelnin et al., 2024) uses noise-injected fine-tuning on salient weights to improve robustness.

While sparse tuning methods are promising, they often require multiple training phases, gradient computations, or architectural modifications to identify important weights. At the same time, the possibility of *combining sparse and low-rank updates* has remained underexplored. One exception is SLTrain (Han et al., 2024), which unifies sparse and low-rank adaptation in a pretraining setting. In the PEFT context, RoSA (Nikdan et al., 2024) also integrates both types of updates. Notably,

our work arrives at this hybrid approach independently, based on a different set of motivations and techniques.

In this paper, we introduce Super, a new sparse PEFT method based on the idea of outlier weights. Inspired by pruning techniques such as WANDA (Sun et al., 2023), we identify and fine-tune only a small set of super weights—outlier weights that have disproportionate impact on model behavior and perplexity. Unlike previous sparse PEFT approaches, Super does not require additional gradient computations or training stages for weight selection. Instead, it relies on a simple, interpretable, training-free metric.

We further propose Supra, a hybrid method that combines Super with LoRA to jointly leverage sparse and low-rank adaptation. Supra brings together the parameter efficiency of Super and the representational power of LoRA, resulting in a flexible and effective fine-tuning strategy. Our extensive experiments on LLMs and downstream tasks demonstrate that both Super and Supra achieve performance on par with or better than existing PEFT methods, with fewer trainable parameters and lower memory consumption.

We summarize our key contributions as follows:

- We propose Super, a new outlier-based sparse fine-tuning method that updates only a small set of important weights, selected without training or gradient statistics.
- We introduce Supra, a hybrid PEFT strategy that combines Super with LoRA, effectively merging sparse and low-rank adaptation.
- We propose a simple yet efficient strategy of setting adaptive rank for LoRA by fixing the number of trainable parameters for every linear layer.
- We demonstrate the effectiveness of our methods across multiple LLMs and downstream tasks, showing that Super and Supra achieve comparable or superior performance to existing PEFT methods with significantly fewer trainable parameters.

Our findings show that *outlier-aware sparse fine-tuning* and *hybrid adaptation* offer promising directions for scalable and effective LLM adaptation.

## 1.1 NOTATION

All key notation used in this paper is summarized in a tabular form in Section A; see Table 3.

## 2 RELATED WORK

### 2.1 PARAMETER–EFFICIENT FINE-TUNING (PEFT)

The prohibitive compute and memory cost of updating all parameters of a large language model (LLM) has motivated a rich literature on *parameter–efficient fine-tuning* (PEFT). Classical adapter-style methods insert small *dense* modules into each Transformer block (e.g., adapters, prefix-tuning, and LoRA), but still back-propagate through the full network, limiting their scalability. Recent work has therefore shifted towards *sparse* or otherwise *structured* updates that explicitly select a subset of parameters to train.

### 2.2 SPARSE FINE-TUNING

Several papers demonstrate that high accuracy can be recovered—even on reasoning-heavy tasks—when a model is updated on only a small fraction of its weights. (Song et al., 2023) analyse the PAC-Bayesian generalisation bound of PEFT and propose SIFT, a gradient-based algorithm that activates at most $k$ parameters per layer during training. (Ansell et al., 2024) show that dynamically grown–pruned sparse deltas scale to 13-B-parameter LLaMA-2 while retaining memory proportional to the sparsity pattern rather than to model size. Orthogonally, (Ma et al., 2024) exploit the observation that only a minority of neurons fire on any given example and skip the forward/backward pass of inactive neurons to accelerate both continual pre-training and supervised fine-tuning by up to 45%. Building on structured sparsity, (Yang et al., 2024) introduce S²FT, which selects a small set

of attention heads and MLP channels, then co-permutes weight matrices so the selected components form dense sub-matrices that can be trained efficiently with ordinary GEMM kernels.

## 2.3 HYBRID LOW-RANK & SPARSE METHODS

A complementary line of work seeks to combine the representational benefits of low-rank adaptation with the compactness of sparsity. (Nikdan et al., 2024) decompose the update into a low-rank adapter plus a very sparse residual and optimise both jointly, delivering full-fine-tuning accuracy at LoRA-sized budgets. For the pre-training regime, (Han et al., 2024) factorise each linear layer into a low-rank term and a *fixed-support* sparse term, achieving up to 73% memory savings while matching full-rank performance.

## 2.4 FINE-TUNING AT THE NEURON OR WEIGHT LEVEL

Moving to finer granularity, (Xu et al., 2024) supervise the updates at the level of individual neurons (NEFT), explicitly identifying task-relevant neurons whose small subset updates suffice to outperform full-parameter fine-tuning. (Zhelnin et al., 2024) extend the idea of salient-weight selection with GIFT-SW, which injects Gaussian noise into *non-salient* columns while learning only the salient ones, closing the gap to full fine-tuning under the same compute budget.

## 2.5 CONNECTIONS TO PRUNING AND OUTLIER-AWARE UPDATES

Our method is inspired by pruning research that measures weight saliency $S_{kq}$ for every weight $W_{kq}$ from a linear layer $W \in \mathbb{R}^{c \times b}$ before deleting parameters. The WANDA metric multiplies a weight's magnitude by the norm of its input activation:

$$S_{kq} = \left( |W_{kq}| \|X_{q:}\|_2 \right)^2,$$ (1)

where $X_{q:} \in \mathbb{R}^{1 \times a}$ is the $q^{\text{th}}$ row of the input matrix matrix $X \in \mathbb{R}^{b \times a}$.

The metric $S_{kq}$ is then used in order to approximate output-level importance and enables one-shot pruning without weight updates (Sun et al., 2023). Instead of discarding low-metric weights, we take the complementary view: we *retain and fine-tune only the outliers*—the high-Wanda-score weights that pruning finds indispensable. This choice is motivated by empirical evidence that perturbing or removing these outliers dramatically degrades perplexity, implying that updating them should yield the largest quality gains for a given parameter budget. In contrast to prior sparse PEFT work, which selects parameters heuristically or grows them during training, our approach uses a pruning-derived saliency signal that is (i) model-specific and (ii) computable in a single forward pass, making it attractive for large-scale deployment.

# 3 ADAPTATION OF LARGE LANGUAGE MODELS

## 3.1 NOTATION

Let $\mathcal{N}$ denote a pre-trained Large Language Model (LLM), and let $\mathcal{W} = \{W_1, W_2, \ldots, W_k\}$ be the collection of all fully connected weight matrices in $\mathcal{N}$, including those within sub-attention layers, with each $W_i \in \mathbb{R}^{c_i \times b_i}$ for $1 \leq i \leq k$. Let the vector $\bar{w} \in \mathbb{R}^{\bar{d}}$ represent the remaining parameters of $\mathcal{N}$, such as biases and normalization parameters, concatenated into a single vector. Given a dataset $\mathcal{D}$ and a loss function $\mathcal{L}(\mathcal{D}; \mathcal{W}, \bar{w})$, the full fine-tuning (FFT) of $\mathcal{N}$ can be formulated as the following optimization problem:

$$\min_{\mathcal{W}, \bar{w}} \mathcal{L}(\mathcal{D}; \mathcal{W}, \bar{w}).$$ (2)

Due to the large scale of modern LLMs – often comprising billions of parameters-performing FFT is computationally expensive and memory-intensive, making it impractical on standard GPUs. A practical alternative is to apply lightweight modifications known as *adapters*, which we formalize next.

Let $\Delta = \{\Delta_1, \Delta_2, \ldots, \Delta_k\}$ denote perturbations applied to the fully connected weights, with $\Delta_i \in \mathbb{R}^{c_i \times b_i}$ for all $i$. Define the adapted weights as $\mathcal{W} + \Delta = \{W_1 + \Delta_1, W_2 + \Delta_2, \ldots, W_k + \Delta_k\}$,

and let $\bar{\delta} \in \mathbb{R}^{\bar{d}}$ represent perturbations to $\bar{w}$. The adapted model is then obtained by solving:

$$\min_{\Delta,\bar{\delta}} \mathcal{L}(\mathcal{D}; \mathcal{W} + \Delta, \bar{w} + \bar{\delta}) \quad \text{s.t.} \quad \mathcal{C}(\Delta, \bar{\delta}), \tag{3}$$

where $\mathcal{C}(\Delta, \bar{\delta})$ encodes constraints on the perturbations (e.g., low-rank or sparse structure) to reduce memory and computational overhead. Notably, if no constraints are imposed, this setting reduces to standard FFT.

In this work, we focus on the common scenario where $\bar{\delta} = 0$, though in principle $\bar{w}$ can also be fine-tuned, especially given its relatively small size compared to $\mathcal{W}$. Additionally, although we assume adaptation of all fully connected weights, our method applies to partial adaptation as well.

**LoRA: Low-Rank Adaptation.** LoRA Hu et al. (2021) constrains each perturbation $\Delta_i$ to be low-rank, specifically:

$$\min_{\Delta} \mathcal{L}(\mathcal{D}; \mathcal{W} + \Delta, \bar{w}) \quad \text{s.t.} \quad \text{rank}(\Delta_i) \leq r \quad \forall i. \tag{4}$$

This reduces the number of trainable parameters for each layer $i$ from $c_i b_i$ to $r(c_i + b_i)$, making fine-tuning more memory-efficient.

**SpA: Sparse Adaptation.** Sparse adaptation (SpA) Sung et al. (2021) enforces sparsity in the perturbations:

$$\min_{\Delta} \mathcal{L}(\mathcal{D}; \mathcal{W} + \Delta, \bar{w}) \quad \text{s.t.} \quad \|\Delta_i\|_0 \leq p c_i b_i \quad \forall i, \tag{5}$$

where $p \in (0, 1]$ is the sparsity density and $\|\cdot\|_0$ denotes the $\ell_0$ pseudo-norm. It is common to fix the sparse support during training, thereby reducing the number of trainable parameters by a factor of $p$.

### 3.2 Super: Mask Generation Algorithm for SpA.

We introduce a mapping $\psi_X : \mathbb{R}^{c \times b} \to \{0, 1\}^{c \times b}$ that takes a matrix of weights $W$ and generates a corresponding fine-tuning mask. For each weight $W_{ij}$, it evaluates the metric $|W_{ij}| \|X_{j:}\|_2$ and selects the $s$ entries with the highest scores. The output is a binary mask with ones at these selected positions:

$$\psi_X(W, s) := \text{mask selecting the } s \text{ weights } W_{ij} \text{ with} \tag{6}$$
$$\text{the highest } |W_{ij}| \|X_{j:}|_2 \text{ values.}$$

Refer to Section B for additional examples.

Our algorithm for generation of sparse masks involves the following procedure:

---
**Algorithm 1** SpA mask generation by Wanda

---
1: **Initialization:** Input matrix $X \in \mathbb{R}^{b \times a}$, weight matrix $W \in \mathbb{R}^{c \times b}$, number of outliers to select $s \in \{0, \cdots, cb\}$.
2: $M \leftarrow \psi_X(W, s)$

---

Unlike many existing methods, our approach does not require a complex or computationally intensive procedure for selecting the trainable parameters of the sparse adapter. Instead, we adopt a simple and highly efficient strategy that can be executed within seconds, even for large-scale models and substantial batch sizes. We refer to this method as **Super**—short for *Selective Update of Parameters via Extreme Ranking*.

The core idea behind Super is to leverage the Wanda metric, which estimates the importance of each parameter by quantifying its impact on the layer-wise $\ell_2$ reconstruction loss upon removal. Specifically, parameters associated with a high reconstruction error are deemed more critical to the model's performance. Thus, Wanda provides a cheap yet effective approximation of parameter importance.

Our central hypothesis is that training should be concentrated on the most impactful parameters to achieve efficient adaptation and rapid convergence. By selecting the top-ranked outlier parameters according to the Wanda metric, Super provides a principled and scalable method for sparse fine-tuning that aligns well with both theoretical intuition and empirical effectiveness.

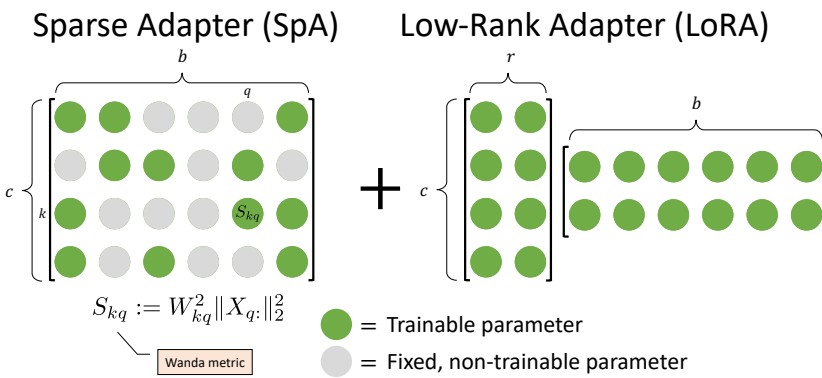

Figure 1: Supra combines a sparse adapter—based on Super weights with a low-rank adapter. To select the parameters for training in the sparse adapter, we employ the WANDA metric (Equation 1). The rank $r$ of the low-rank adapter is determined dynamically using Equation 7.

### 3.3 SUPRA: ADAPTIVE LOW-RANK AND SPARSE ADAPTATION.

Supra is a hybrid fine-tuning strategy that combines two complementary approaches: a sparse adapter selected via the **Super** strategy and a low-rank adapter based on **LoRA**. This unified framework is designed to maximize the strengths of both methods: the stability and selectivity of sparse adaptation, and the flexibility and expressiveness of low-rank adaptation.

The intuition behind this design stems from the observation that LoRA, by construction, learns updates in a low-dimensional subspace. Specifically, although the LoRA adapter can be interpreted as a full matrix when expanded, the effective number of degrees of freedom is limited by the chosen rank. Consequently, LoRA dynamically explores and updates a low-rank subspace of parameters throughout fine-tuning. This evolving subspace allows the model to gradually adjust its internal representations and better align with the target task.

In contrast, sparse adaptation—particularly when guided by Super—selects a fixed subset of important parameters to train. These parameters are chosen based on their estimated influence on the model's output, as measured by the Wanda metric. Once selected, this subset remains constant during training, resulting in updates restricted to a fixed subspace.

By combining these two adapters, Supra achieves a powerful balance: the sparse adapter ensures that the most crucial parameters from the pre-trained model are retained and further optimized, preserving core knowledge, while the LoRA component dynamically adapts to the target task by learning task-specific directions in the parameter space. In essence:

- The sparse adapter captures and preserves *essential knowledge* embedded in the original model, focusing on parameters that were previously influential.

- The low-rank adapter enables *task-specific generalization* by learning new subspaces of importance in a structured and efficient manner.

This synergy allows Supra to fine-tune large language models both accurately and efficiently, achieving fast convergence while minimizing memory and compute overhead. Moreover, the explicit control over the parameter budget and its division between sparse and low-rank components enables practitioners to tailor the method to different resource constraints or adaptation scenarios.

### 3.4 ADAPTIVE BUDGET ALLOCATION IN SUPRA

Supra employs an adaptive strategy to determine the rank $r_i$ of the Low-Rank adapter based on a user-defined parameter $p \in (0, 1]$, representing the overall sparsity density.

Rather than fixing the rank for each linear layer $W_i$, we fix the total number of trainable parameters per layer. Specifically, for a linear layer $W_i \in \mathbb{R}^{c_i \times b_i}$, we allocate a total budget of $\lfloor pc_i b_i \rfloor$ trainable

parameters. To divide this budget between the Low-Rank and Sparse adapters, we introduce a new parameter $\alpha \in [0, 1]$, referred to as the *Low-Rank parameters ratio*.

- If $\alpha = 1$, the entire budget is allocated to the Low-Rank adapter, recovering the standard LoRA setup.
- If $\alpha = 0$, the full budget is allocated to the Sparse adapter, resulting in standard Sparse Adaptation.
- If $\alpha = 0.5$, the budget is evenly split between Low-Rank and Sparse adapters.

Under this scheme, the Low-Rank adapter receives $\lfloor \alpha pc_i b_i \rfloor$ trainable parameters, while the Sparse adapter is assigned $\lfloor (1 - \alpha)pc_i b_i \rfloor$.

Unlike sparse adaptation, which directly operates on a fixed number of parameters, Low-Rank adaptation requires specifying a rank $r_i$. To bridge this gap, we define the following adaptive rule for selecting $r_i$ based on $\alpha$ and $p$:

$$r_i = \left\lceil \frac{\alpha pc_i b_i}{c_i + b_i} \right\rceil. \tag{7}$$

This formulation ensures that the Low-Rank adapter uses $l_i = r_i(c_i + b_i) = \left\lceil \frac{\alpha pc_i b_i}{c_i + b_i} \right\rceil (c_i + b_i)$ parameters, and the Sparse adapter is assigned the remaining $s_i = \lfloor pc_i b_i \rfloor - l_i$ parameters, so the total number of trainable parameters remains consistent: $s_i + l_i = \lfloor pc_i b_i \rfloor$.

We note that due to rounding in equation 7, the actual ratio of Low-Rank to total trainable parameters may deviate slightly from the target value $\alpha$. In particular, $\frac{l_i}{l_i + s_i} \neq \alpha$. This discrepancy arises from the ceiling operation in equation 7, which ensures that $r_i$ remains an integer. However, we find that such deviations are minor in practice, and the value of $\alpha$ can be adjusted to compensate for rounding effects during empirical tuning.

## 4 EXPERIMENTS

### 4.1 EXPERIMENTAL SETUP

**Datasets** All experiments in this work are conducted on the **Math10K** dataset, originally introduced in the LLM-Adapters study by Hu et al. (2023). This dataset is specifically curated for fine-tuning and evaluating language models on math reasoning tasks.

The Math10K dataset consists of 10,000 high-quality arithmetic reasoning problems drawn from multiple publicly available benchmarks, including GSM8K (Cobbe et al., 2021), AQuA (Ling et al., 2017), MAWPS and MAWPS-Single (Koncel-Kedziorski et al., 2016). The dataset was constructed by selecting high-quality samples that provide both equations and answers, and then augmenting them with step-by-step rationales generated by ChatGPT using zero-shot chain-of-thought prompting. Only samples with correct answers were retained to ensure data quality.

For evaluation, we follow the protocol of Hu et al. (2023) and report accuracy scores on six arithmetic reasoning benchmarks:

- **GSM8K** (Cobbe et al., 2021): Grade-school math word problems with diverse linguistic structure.
- **SVAMP** (Patel et al., 2021): One-variable arithmetic problems derived via minimal changes from an existing dataset.
- **MultiArith** (Roy & Roth, 2016): Multi-step arithmetic problems requiring multiple operations.
- **AddSub** (Hosseini et al., 2014): Basic addition and subtraction problems.
- **AQuA** (Ling et al., 2017): Algebraic word problems with answer options and natural language rationales.
- **SingleEq** (Koncel-Kedziorski et al., 2015): Algebra word problems that map to single-variable equations of varying complexity.

These datasets collectively assess the model's ability to reason over arithmetic structures and solve problems requiring symbolic manipulation and multi-step inference.

**Training Details** All models are fine-tuned using adapter-based methods, where only a small subset of parameters is trained while the rest of the model remains frozen. We compare several adaptation strategies, including LoRA, SIFT and our proposed methods Super and Supra.

**Parameter Budgets.** We conduct experiments under two different parameter budgets, defined by the LoRA rank: `lora_r` = 8 (moderate budget) and `lora_r` = 4 (low budget). For Supra, we vary the parameter split ratio $\alpha \in \{0.3, 0.55, 0.8\}$ to evaluate the trade-off between sparse and low-rank adaptation. All methods use the same underlying backbone model - LLaMA-3 (1B/3B/8B) (Dubey et al., 2024), and all hyperparameters are kept consistent across runs for fair comparison.

**Evaluation Metric.** We report average accuracy along with standard deviation across multiple random seeds. Each dataset is evaluated independently, and the overall average accuracy is used as a summary metric to compare different methods.

**Reproducibility.** Following the LLM-Adapters setup, all experiments are conducted using publicly available datasets and reproducible fine-tuning pipelines. The Math10K dataset and associated prompt templates used to generate rationales with ChatGPT are detailed in Hu et al. (2023), Appendix A.

For all our experiments we used a single Nvidia A100 GPU (Choquette et al., 2021).

## 4.2 COMPARISON WITH OTHER FINE-TUNING METHODS

We evaluate Supra and other baselines on six mathematical reasoning benchmarks: MultiArith, GSM8K, AddSub, AQuA, SingleEq, and SVAMP. The results under two different fine-tuning budgets—defined by the LoRA rank $r \in \{8, 4\}$ – are shown in Tables 1 and 2.

**High Budget Setting (`lora_r` = 8).** Under a relatively generous parameter budget, the hybrid Supra method consistently achieves the best or near-best results across most tasks. In particular, Supra with $\alpha = 0.8$, which dedicates the majority of its budget to the Low-Rank adapter, achieves the highest average accuracy of **60.6%**, outperforming all baselines including LoRA (58.7%) and Super (59.2%). This highlights the advantage of combining dynamic low-rank updates with a fixed set of high-importance sparse parameters. Interestingly, LoRA performs well in this setting, reflecting the benefits of learning a task-specific subspace when sufficient capacity is available.

**Low Budget Setting (`lora_r` = 4).** When the rank is reduced to 4, the overall number of trainable parameters drops substantially, making the selection of which parameters to train far more critical. In this constrained regime, the **Super** method – using only the sparse adapter based on outlier parameter selection achieves the best average accuracy of **58.2%**, outperforming all other methods, including Supra variants and LoRA. This suggests that when budget is tight, focusing on the most influential parameters (as estimated by the Wanda metric) is more beneficial than attempting to learn a low-rank subspace.

Supra remains competitive in this regime, with the best-performing variant ($\alpha = 0.3$) reaching 58.0% average accuracy, slightly below Super but still ahead of LoRA (57.0%) and SIFT (55.3%). However, as the value of $\alpha$ increases (i.e., shifting more budget to LoRA), performance tends to degrade, confirming that low-rank updates alone are insufficient when the parameter budget is too limited.

These findings offer several important insights:

- **Importance of parameter selection under constraints.** In low-budget settings, it is crucial to carefully select which parameters to train. The Super adapter achieves this via a principled selection of outlier weights, leading to robust performance even with minimal updates.
- **Subspace adaptivity vs. parameter criticality.** LoRA excels when there is enough capacity to explore new subspaces dynamically. However, in low-capacity regimes, maintaining a fixed set of critical parameters as in Super is more effective.

- **Supra as a unifying strategy.** Supra benefits from both perspectives: it preserves essential knowledge from the pre-trained model via sparse adaptation, and simultaneously adapts to the target task by learning a low-rank subspace. The ability to balance these via the $\alpha$ parameter allows Supra to operate effectively across a range of parameter budgets.

Overall, these results validate our design of Supra as a flexible and powerful fine-tuning method, capable of adapting to both high and low training budgets by interpolating between sparse and low-rank regimes.

Table 1: Comparison of methods with `lora_r = 8`. Results are accuracy (%) $\pm$ standard deviation.

| Method | MultiArith | GSM8K | AddSub | AQuA | SingleEq | SVAMP | Average |
|---|---|---|---|---|---|---|---|
| LoRA | $92.4 \pm 3.3$ | $23.4 \pm 2.1$ | $86.5 \pm 2.6$ | $\mathbf{25.5 \pm 1.6}$ | $83.5 \pm 2.1$ | $45.6 \pm 2.4$ | $58.7 \pm 1.3$ |
| SIFT | $92.2 \pm 1.7$ | $23.1 \pm 1.2$ | $83.1 \pm 1.0$ | $23.1 \pm 1.9$ | $82.0 \pm 1.2$ | $44.2 \pm 0.7$ | $57.1 \pm 0.5$ |
| SIFT (rand) | $87.8 \pm 1.8$ | $23.9 \pm 1.0$ | $84.0 \pm 4.3$ | $21.8 \pm 0.8$ | $83.5 \pm 1.9$ | $48.1 \pm 2.3$ | $57.4 \pm 0.3$ |
| Rosa | $91.7 \pm 0.8$ | $23.5 \pm 1.1$ | $84.0 \pm 0.9$ | $21.3 \pm 0.4$ | $82.4 \pm 1.3$ | $44.6 \pm 1.7$ | $57.9 \pm 0.2$ |
| Super | $91.6 \pm 1.1$ | $25.4 \pm 0.9$ | $86.3 \pm 3.2$ | $22.9 \pm 1.0$ | $84.6 \pm 1.7$ | $46.7 \pm 0.9$ | $59.2 \pm 0.3$ |
| Supra (0.3) | $93.6 \pm 0.5$ | $25.8 \pm 0.7$ | $87.0 \pm 0.5$ | $25.1 \pm 1.8$ | $85.7 \pm 0.7$ | $47.5 \pm 1.4$ | $60.4 \pm 0.6$ |
| Supra (0.55) | $94.8 \pm 1.8$ | $25.9 \pm 1.4$ | $87.1 \pm 1.2$ | $23.8 \pm 0.9$ | $86.5 \pm 0.7$ | $47.1 \pm 3.6$ | $60.6 \pm 1.2$ |
| **Supra (0.8)** | $\mathbf{95.1 \pm 1.5}$ | $\mathbf{26.0 \pm 0.6}$ | $\mathbf{88.5 \pm 1.0}$ | $22.7 \pm 1.1$ | $\mathbf{87.2 \pm 0.9}$ | $\mathbf{48.3 \pm 1.5}$ | $\mathbf{60.6 \pm 0.3}$ |

Table 2: Comparison of methods with `lora_r = 4`. Results are accuracy (%) $\pm$ standard deviation.

| Method | MultiArith | GSM8K | AddSub | AQuA | SingleEq | SVAMP | Average |
|---|---|---|---|---|---|---|---|
| LoRA | $90.1 \pm 2.9$ | $24.4 \pm 1.4$ | $83.7 \pm 3.3$ | $23.1 \pm 2.7$ | $81.3 \pm 2.9$ | $43.6 \pm 4.4$ | $57.0 \pm 1.7$ |
| SIFT | $88.3 \pm 2.2$ | $21.1 \pm 1.8$ | $80.9 \pm 1.7$ | $22.2 \pm 2.2$ | $78.9 \pm 1.4$ | $41.5 \pm 1.2$ | $55.3 \pm 1.1$ |
| Super | $88.9 \pm 0.9$ | $23.5 \pm 1.5$ | $\mathbf{85.5 \pm 2.0}$ | $24.3 \pm 1.8$ | $\mathbf{84.3 \pm 1.2}$ | $44.5 \pm 1.8$ | $\mathbf{58.2 \pm 0.5}$ |
| SIFT (rand) | $87.1 \pm 3.2$ | $23.4 \pm 0.6$ | $83.8 \pm 2.0$ | $21.1 \pm 1.5$ | $82.2 \pm 1.1$ | $\mathbf{46.8 \pm 2.4}$ | $57.4 \pm 0.7$ |
| Rosa | $83.7 \pm 2.6$ | $21.9 \pm 0.3$ | $81.6 \pm 3.3$ | $21.2 \pm 1.4$ | $78.9 \pm 1.1$ | $46.3 \pm 2.1$ | $55.6 \pm 1.3$ |
| Supra (0.3) | $92.5 \pm 3.0$ | $24.8 \pm 1.1$ | $82.4 \pm 2.1$ | $23.6 \pm 1.0$ | $82.7 \pm 1.1$ | $42.7 \pm 1.6$ | $58.0 \pm 0.7$ |
| Supra (0.55) | $\mathbf{93.1 \pm 1.8}$ | $24.1 \pm 0.5$ | $82.9 \pm 2.8$ | $22.7 \pm 2.8$ | $83.5 \pm 0.9$ | $43.8 \pm 1.3$ | $57.8 \pm 1.1$ |
| Supra (0.8) | $91.7 \pm 0.7$ | $22.2 \pm 1.8$ | $81.6 \pm 3.0$ | $\mathbf{24.8 \pm 2.8}$ | $82.5 \pm 1.4$ | $43.2 \pm 1.1$ | $57.2 \pm 0.6$ |

Additional experiments and more comparisons with other methods can be found in Section C.

## 5 DISCUSSION

Our work demonstrates that fine-tuning large language models can be significantly improved through careful selection and allocation of trainable parameters. In particular, we introduced two key innovations: (1) an adaptive strategy for setting the rank of Low-Rank adapters by fixing the number of trainable parameters per layer, and (2) a simple yet effective method for selecting sparse adaptation parameters based on outlier scores derived from the WANDA metric. These contributions enabled us to design Super, a sparse fine-tuning method that requires no additional training or gradient information, and Supra, a hybrid strategy that combines sparse and low-rank adaptation into a unified framework. The adaptive rank strategy ensures that the low-rank component scales naturally with the layer size, while the use of WANDA outliers provides an interpretable and training-free mechanism to identify impactful weights for sparse updates. Together, these methods achieve strong performance with minimal parameter overhead across multiple LLMs and tasks. Importantly, our approach remains simple to implement and incurs little computational overhead, making it well-suited for scalable and practical deployment in real-world scenarios. This highlights the potential of combining principled, interpretable metrics with efficient parameter allocation to push the boundaries of parameter-efficient fine-tuning.

## 6 LIMITATIONS

Despite the empirical gains documented in Section 4, our study has several limitations that we hope future work will address.

**Scope of evaluation.** All experiments are confined to the **Math10K** benchmark and six arithmetic–reasoning test suites. While this domain is attractive for its controlled difficulty and clear accuracy metric, it represents a narrow slice of the broad application space of LLMs. In particular, language–generation tasks that hinge on open-ended semantics (e.g., summarisation, dialogue, or code synthesis) may stress different model components than symbolic math reasoning; the relative benefits of sparse, low-rank, and hybrid updates could therefore shift in other settings. A more comprehensive assessment across diverse modalities, instruction-following tasks, and safety-critical benchmarks is necessary before drawing general conclusions.

**Dependence on the *WANDA* saliency proxy.** SUPER (and hence SUPRA) selects trainable weights using an activation-weighted magnitude heuristic inherited from the WANDA pruning metric. Although this choice is fast and training-free, it is still an *approximation* of true importance; weights that appear non-salient under the proxy may in fact become critical once the task distribution shifts or once other parameters are updated. We do not study how sensitive performance is to mis-rankings, nor whether more expressive but costlier criteria (e.g. curvature-aware saliency or gradient-flow statistics) would yield better sparse masks.

**Fixed sparse support.** The sparse component of SUPRA is *static* throughout fine-tuning: once a weight is deemed non-trainable, it can never be activated. Dynamic sparsity—allowing the mask to grow, prune, or re-allocate budget during training—has shown promise in other contexts and may close the gap to full fine-tuning in edge cases where the initial selection is sub-optimal.

**Layer-local parameter budgets.** Our adaptive rank rule (Equation (7)) allocates the same *fraction* of parameters to every linear layer. In practice, different layers contribute unequally to downstream task performance; early attention blocks, for example, often admit more aggressive compression than middle MLP layers. A global, data-driven re-allocation of the parameter budget—akin to neural architecture search—might improve efficiency further.

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

## A    TABLE OF FREQUENTLY USED NOTATION

Table 3: Notation table

| | | |
|---|---|---|
| $W$ | – | One linear layer (matrix) of LLM, $W \in \mathbb{R}^{c \times b}$. |
| $p$ | – | Sparsity density. Represents the ratio of parameters for fine-tuning. $p \in (0, 1]$. |
| $\alpha$ | – | Low-Rank parameters ratio. The ratio of Low-Rank parameters (for Supra). $\alpha \in [0, 1]$. |
| $l$ | – | Number of Low-Rank trainable parameters. $l \in \{0, \cdots, cb\}$. |
| $s$ | – | Number of Sparse Adapter trainable parameters. $s \in \{0, \cdots, cb\}$. |
| $i, j$ | – | Matrix indexes for rows and columns respectively. |
| $k, q$ | – | Matrix indexes for row and column for pruning respectively. |
| $x_j$ | – | $j^{\text{th}}$ element of the vector $x$. |
| $e^j$ | – | Unit vector with 1 at the $j^{\text{th}}$ position and 0 everywhere else. $e^j \in \mathbb{R}^{b \times 1}$. |
| $W_{ij}$ | – | one entry of the matrix $W$ on the intersection of the $i^{\text{th}}$ row and $j^{\text{th}}$ column. |
| $W_{i:}$ | – | $i^{\text{th}}$ row of the matrix $W$, $W_{i:}$ is a row-vector. |
| $W_{:j}$ | – | $j^{\text{th}}$ column of the matrix $W$, $W_{:j}$ is a column-vector. |
| $W_{i,j_1:j_2}$ | $:=$ | $(W_{ij_1} \cdots W_{ij_2}) \in \mathbb{R}^{1 \times (j_2 - j_1)}, \quad 1 \le j_1 \le j_2 \le b$. |
| $W_{i_1:i_2,j}$ | $:=$ | $(W_{i_1j} \cdots W_{i_2j})^\top \in \mathbb{R}^{(i_2 - i_1) \times 1}, \quad 1 \le i_1 \le i_2 \le c$. |
| $W_{i_1:i_2,j_1:j_2}$ | $:=$ | $(W_{i_1:i_2,j_1} \cdots W_{i_1:i_2,j_2}) \in \mathbb{R}^{(i_2 - i_1) \times (j_2 - j_1)}, \quad 1 \le j_1 \le j_2 \le b, \quad 1 \le i_1 \le i_2 \le c$. |
| $W_{i:,j:}$ | $:=$ | $(W_{i:c,j} \cdots W_{i:c,b}) \in \mathbb{R}^{(c-i) \times (b-j)}, \quad 1 \le j \le b, \quad 1 \le i \le c$. |
| $X$ | – | Input matrix for a linear layer $W$, $X \in \mathbb{R}^{b \times a}$. |
| $\widehat{W}$ | – | One linear layer after fine tuning weights from $W$, $\widehat{W} \in \mathbb{R}^{c \times b}$. |
| $\Delta$ | $:=$ | $\widehat{W} - W$ – change of the weight matrix (before and after fine tuning), $\Delta \in \mathbb{R}^{c \times b}$. |
| $p$ | $:=$ | $\frac{1}{cb}\|M\|_F^2$ – sparsity ratio $p \in [0, 1)$. |
| $X^l$ | – | $l^{\text{th}}$ calibration sample, $X^l \in \mathbb{R}^{b \times a}$. |
| $1 \in \mathbb{R}^{c \times b}$ | – | The matrix with size $c \times b$, all entries of which are equal to 1. |
| $0 \in \mathbb{R}^{c \times b}$ | – | The matrix with size $c \times b$, all entries of which are equal to 0. |
| $\|x\|_2$ | $:=$ | $\sqrt{\sum_{i=1} |x_i|^2}$ – $l^2$-norm of vector. |
| $\|A\|_F$ | $:=$ | $\sqrt{\sum_{i=1} \sum_{j=1} |A_{ij}|^2}$ – Frobenius norm of the matrix. |
| $\psi_X(W, r)$ | $:=$ | mask of the $r$ weights $W_{ij}$ with the smallest $|W_{ij}| \|X_{j:}\|_2$ values. |
| $\phi(h)$ | $:=$ | vector of indices of non-zero elements from $h$. |
| $M$ | – | Fine tuning mask. Equal to one only for entries of $W$ that should be trained while fine tuning. |
| $\odot$ | – | Element-wise product of two matrices (Hadamard product). |
| $\otimes$ | – | Outer product of two vectors. |
| $\lfloor x \rfloor$ | $:=$ | $\max\{n \in \mathbb{Z} \mid n \le x\}$ – floor function. |
| $\lceil x \rceil$ | $:=$ | $\min\{n \in \mathbb{Z} \mid n \ge x\}$ – ceiling function. |

## B    WANDA METRIC FOR GENERATING FINE-TUNING MASK

We introduce a mapping $\psi_X : \mathbb{R}^{c \times b} \to \{0, 1\}^{c \times b}$ that takes a matrix of weights $W$ and generates a corresponding fine-tuning mask. For each weight $W_{ij}$, it evaluates the metric $|W_{ij}| \|X_{j:}\|_2$ and selects the $r$ entries with the highest scores. The output is a binary mask with ones at these selected positions:

$$\psi_X(W, r) := \text{mask selecting the } r \text{ weights } W_{ij} \text{ with} \tag{8}$$
$$\text{the highest } |W_{ij}| |X_{j:}|_2 \text{ values.}$$

For example, let us have a weight matrix

$$W = \begin{pmatrix} 3 & -2 \\ -2 & 4 \\ 1 & -6 \end{pmatrix},$$

and an input matrix

$$X = \begin{pmatrix} 4 & 3 \\ 0 & 1 \end{pmatrix},$$

now we compute the metric $|W_{ij}|\|X_{j:}\|_2$ for every entry of $W$:

$$\begin{pmatrix} |W_{11}|\|X_{1:}\|_2 & |W_{12}|\|X_{2:}\|_2 \\ |W_{21}|\|X_{1:}\|_2 & |W_{22}|\|X_{2:}\|_2 \\ |W_{31}|\|X_{1:}\|_2 & |W_{32}|\|X_{2:}\|_2 \end{pmatrix} = \begin{pmatrix} 3 \cdot 5 & 2 \cdot 1 \\ 2 \cdot 5 & 4 \cdot 1 \\ 1 \cdot 5 & 6 \cdot 1 \end{pmatrix} = \begin{pmatrix} 15 & 2 \\ 10 & 4 \\ 5 & 6 \end{pmatrix},$$

then

$$\psi_X(W, 1) = \begin{pmatrix} 1 & 0 \\ 0 & 0 \\ 0 & 0 \end{pmatrix}, \quad \psi_X(W, 2) = \begin{pmatrix} 1 & 0 \\ 1 & 0 \\ 0 & 0 \end{pmatrix}, \quad \psi_X(W, 4) = \begin{pmatrix} 1 & 0 \\ 1 & 0 \\ 1 & 1 \end{pmatrix}.$$

## C  ADDITIONAL EXPERIMENTS

## D  EXPERIMENTS WITH DIFFERENT LEARNING RATES

### D.1  LORA

Table 4: Performance of LoRA with `lora_r = 4` across different learning rates. Results are accuracy (%) ± standard deviation.

| Learning Rate | MultiArith | GSM8K | AddSub | AQuA | SingleEq | SVAMP | Average |
|---|---|---|---|---|---|---|---|
| 2e−5 | – | – | – | – | – | – | – |
| 5e−5 | – | – | – | – | – | – | – |
| 1e−4 | – | – | – | – | – | – | – |
| 2e−4 | 89.5 ± 2.4 | 21.6 ± 1.4 | 69.3 ± 3.3 | **23.1 ± 2.7** | 74.1 ± 1.4 | 39.9 ± 4.5 | 52.9 ± 0.9 |
| 5e−4 | **90.1 ± 2.9** | **24.4 ± 1.4** | 78.4 ± 3.5 | 22.5 ± 3.0 | 78.6 ± 4.3 | 40.8 ± 4.2 | 55.8 ± 2.0 |
| 1e−3 | 89.6 ± 3.1 | 22.5 ± 1.3 | **83.7 ± 3.3** | 22.1 ± 2.3 | **81.3 ± 2.9** | 43.0 ± 2.6 | **57.0 ± 1.7** |
| 2e−3 | 87.7 ± 2.8 | 21.3 ± 1.5 | 82.8 ± 3.1 | 22.8 ± 0.6 | 80.5 ± 2.4 | **43.6 ± 4.4** | 56.5 ± 1.6 |
| **Best LR** | 5e−4 | 5e−4 | 1e−3 | 2e−4 | 1e−3 | 2e−3 | 1e−3 |

Table 5: Performance of LoRA with `lora_r = 8` across different learning rates. Results are accuracy (%) ± standard deviation.

| Learning Rate | MultiArith | GSM8K | AddSub | AQuA | SingleEq | SVAMP | Average |
|---|---|---|---|---|---|---|---|
| 2e−5 | 51.6 ± 1.8 | 14.5 ± 0.2 | 61.5 ± 0.0 | 19.2 ± 0.6 | 61.4 ± 1.4 | 37.4 ± 0.2 | 40.9 ± 0.3 |
| 5e−5 | 70.3 ± 2.8 | 18.4 ± 0.3 | 59.7 ± 2.9 | 21.8 ± 0.5 | 64.8 ± 1.6 | 35.7 ± 1.1 | 45.1 ± 0.2 |
| 1e−4 | 78.9 ± 4.0 | 20.2 ± 2.5 | 58.8 ± 1.4 | 25.3 ± 2.0 | 69.0 ± 5.6 | 37.0 ± 4.6 | 48.2 ± 1.6 |
| 2e−4 | 86.9 ± 3.6 | **23.4 ± 2.1** | 70.0 ± 3.0 | 22.7 ± 0.9 | 75.4 ± 3.8 | 40.3 ± 4.4 | 53.1 ± 1.3 |
| 5e−4 | 90.3 ± 9.0 | 22.0 ± 5.6 | 68.2 ± 24.0 | **25.5 ± 1.6** | 68.8 ± 26.3 | 37.0 ± 15.7 | 52.0 ± 13.2 |
| 1e−3 | **92.4 ± 3.3** | 20.9 ± 1.4 | 77.3 ± 9.9 | 23.2 ± 1.7 | 71.8 ± 15.3 | 39.3 ± 11.3 | 54.1 ± 6.2 |
| 2e−3 | 92.3 ± 1.4 | 21.6 ± 0.5 | **86.5 ± 2.6** | 22.6 ± 1.3 | **83.5 ± 2.1** | **45.6 ± 2.4** | **58.7 ± 1.3** |
| **Best LR** | 1e−3 | 2e−4 | 2e−3 | 5e−4 | 2e−3 | 2e−3 | 2e−3 |

Table 6: Performance of LoRA with `lora_r = 16` across different learning rates. Results are accuracy (%) ± standard deviation.

| Learning Rate | MultiArith | GSM8K | AddSub | AQuA | SingleEq | SVAMP | Average |
|---|---|---|---|---|---|---|---|
| 2e−5 | 50.4 ± 2.5 | 15.0 ± 0.1 | 60.1 ± 0.8 | 21.0 ± 2.6 | 61.5 ± 0.6 | 37.8 ± 1.3 | 41.0 ± 0.7 |
| 5e−5 | 68.3 ± 2.1 | 19.0 ± 0.6 | 61.5 ± 2.2 | 21.3 ± 1.6 | 65.0 ± 3.3 | 35.7 ± 1.5 | 45.1 ± 1.2 |
| 1e−4 | 74.6 ± 1.9 | 20.8 ± 0.7 | 60.9 ± 3.5 | **23.5 ± 2.2** | 69.2 ± 5.6 | 36.8 ± 3.2 | 47.6 ± 2.0 |
| 2e−4 | 90.4 ± 2.2 | 23.2 ± 0.7 | 66.3 ± 8.6 | 22.8 ± 1.7 | 72.7 ± 4.7 | 41.1 ± 3.3 | 52.8 ± 3.3 |
| 5e−4 | **93.3 ± 1.6** | **24.8 ± 1.8** | 80.5 ± 2.4 | 22.8 ± 2.9 | 81.0 ± 2.2 | **45.8 ± 4.0** | 58.0 ± 1.1 |
| 1e−3 | 91.4 ± 2.8 | 24.4 ± 4.7 | **84.7 ± 5.5** | 22.0 ± 1.4 | **82.1 ± 5.7** | 43.7 ± 7.9 | **58.1 ± 4.0** |
| 2e−3 | – | – | – | – | – | – | – |
| **Best LR** | 5e−4 | 5e−4 | 1e−3 | 1e−4 | 1e−3 | 5e−4 | 1e−3 |

## D.2 SIFT

Table 7: Performance of SIFT with `lora_r` $= 4$ across different learning rates. Results are accuracy (%) $\pm$ standard deviation.

| Learning Rate | MultiArith | GSM8K | AddSub | AQuA | SingleEq | SVAMP | Average |
|---|---|---|---|---|---|---|---|
| 2e−5 | – | – | – | – | – | – | – |
| 5e−5 | – | – | – | – | – | – | – |
| 1e−4 | – | – | – | – | – | – | – |
| 2e−4 | $74.1 \pm 2.5$ | $19.6 \pm 1.0$ | $69.9 \pm 3.0$ | $21.2 \pm 0.9$ | $75.0 \pm 2.0$ | $40.6 \pm 2.1$ | $50.1 \pm 1.3$ |
| 5e−4 | $80.9 \pm 2.7$ | $\mathbf{21.1 \pm 1.8}$ | $78.5 \pm 0.7$ | $\mathbf{22.2 \pm 2.2}$ | $78.2 \pm 1.6$ | $\mathbf{41.5 \pm 1.2}$ | $53.7 \pm 0.8$ |
| 1e−3 | $88.1 \pm 0.8$ | $20.6 \pm 0.9$ | $\mathbf{80.9 \pm 1.7}$ | $22.0 \pm 3.3$ | $\mathbf{78.9 \pm 1.4}$ | $41.4 \pm 1.6$ | $\mathbf{55.3 \pm 1.1}$ |
| 2e−3 | $\mathbf{88.3 \pm 2.2}$ | $17.0 \pm 0.9$ | $80.6 \pm 2.2$ | $20.4 \pm 2.4$ | $77.5 \pm 3.0$ | $40.1 \pm 2.8$ | $54.0 \pm 1.5$ |
| **Best LR** | 2e−3 | 5e−4 | 1e−3 | 5e−4 | 1e−3 | 5e−4 | 1e−3 |

Table 8: Performance of SIFT with `lora_r` $= 8$ across different learning rates. Results are accuracy (%) $\pm$ standard deviation.

| Learning Rate | MultiArith | GSM8K | AddSub | AQuA | SingleEq | SVAMP | Average |
|---|---|---|---|---|---|---|---|
| 2e−5 | $49.4 \pm 1.1$ | $15.5 \pm 0.3$ | $59.8 \pm 1.7$ | $20.9 \pm 0.7$ | $62.9 \pm 0.7$ | $35.1 \pm 0.7$ | $40.6 \pm 0.6$ |
| 5e−5 | $62.1 \pm 5.4$ | $18.4 \pm 0.3$ | $60.9 \pm 2.3$ | $22.0 \pm 1.4$ | $68.2 \pm 1.1$ | $36.8 \pm 2.0$ | $44.8 \pm 0.8$ |
| 1e−4 | $71.3 \pm 7.7$ | $21.3 \pm 0.6$ | $64.3 \pm 0.3$ | $19.6 \pm 1.3$ | $71.3 \pm 3.5$ | $37.7 \pm 1.4$ | $47.6 \pm 0.6$ |
| 2e−4 | $81.9 \pm 1.8$ | $20.4 \pm 1.2$ | $74.5 \pm 3.3$ | $21.4 \pm 0.6$ | $77.8 \pm 0.3$ | $\mathbf{44.2 \pm 0.7}$ | $53.4 \pm 0.2$ |
| 5e−4 | $88.1 \pm 1.5$ | $\mathbf{23.1 \pm 1.2}$ | $83.1 \pm 2.1$ | $\mathbf{23.1 \pm 1.9}$ | $\mathbf{82.0 \pm 1.2}$ | $43.2 \pm 1.1$ | $\mathbf{57.1 \pm 0.5}$ |
| 1e−3 | $\mathbf{92.2 \pm 1.7}$ | $20.9 \pm 0.7$ | $\mathbf{83.1 \pm 1.0}$ | $21.3 \pm 0.7$ | $82.0 \pm 0.3$ | $42.6 \pm 1.2$ | $57.0 \pm 0.3$ |
| 2e−3 | – | – | – | – | – | – | – |
| **Best LR** | 1e−3 | 5e−4 | 1e−3 | 5e−4 | 5e−4 | 2e−4 | 5e−4 |

Table 9: Performance of SIFT with `lora_r` $= 16$ across different learning rates. Results are accuracy (%) $\pm$ standard deviation.

| Learning Rate | MultiArith | GSM8K | AddSub | AQuA | SingleEq | SVAMP | Average |
|---|---|---|---|---|---|---|---|
| 2e−5 | $56.4 \pm 4.9$ | $17.6 \pm 0.3$ | $63.5 \pm 1.9$ | $21.7 \pm 1.0$ | $67.8 \pm 1.3$ | $38.3 \pm 1.0$ | $44.2 \pm 0.7$ |
| 5e−5 | $74.7 \pm 2.9$ | $20.9 \pm 0.9$ | $68.9 \pm 2.8$ | $19.6 \pm 1.5$ | $74.7 \pm 1.2$ | $41.2 \pm 1.3$ | $50.0 \pm 0.5$ |
| 1e−4 | $80.3 \pm 1.8$ | $22.1 \pm 1.0$ | $74.6 \pm 0.5$ | $20.2 \pm 1.1$ | $78.0 \pm 1.4$ | $43.3 \pm 1.6$ | $53.1 \pm 0.3$ |
| 2e−4 | $84.7 \pm 5.8$ | $\mathbf{24.0 \pm 0.8}$ | $83.2 \pm 1.3$ | $\underline{\mathbf{24.3 \pm 0.8}}$ | $83.3 \pm 0.3$ | $\mathbf{44.8 \pm 1.6}$ | $57.4 \pm 0.9$ |
| 5e−4 | $92.8 \pm 2.1$ | $23.5 \pm 1.5$ | $85.7 \pm 2.1$ | $23.9 \pm 1.5$ | $83.5 \pm 1.7$ | $44.4 \pm 1.5$ | $\mathbf{59.0 \pm 1.1}$ |
| 1e−3 | $\underline{\mathbf{93.5 \pm 1.4}}$ | $20.8 \pm 0.5$ | $\underline{\mathbf{87.0 \pm 0.7}}$ | $21.1 \pm 0.5$ | $\mathbf{84.1 \pm 1.2}$ | $42.0 \pm 1.1$ | $58.1 \pm 0.1$ |
| 2e−3 | – | – | – | – | – | – | – |
| **Best LR** | 1e−3 | 2e−4 | 1e−3 | 2e−4 | 1e−3 | 2e−4 | 5e−4 |

## D.3 SIFT RANDOM

Table 10: Performance of SIFT_random with `lora_r` = 4 across different learning rates. Results are accuracy (%) ± standard deviation.

| Learning Rate | MultiArith | GSM8K | AddSub | AQuA | SingleEq | SVAMP | Average |
|---|---|---|---|---|---|---|---|
| 2e−5 | – | – | – | – | – | – | – |
| 5e−5 | – | – | – | – | – | – | – |
| 1e−4 | – | – | – | – | – | – | – |
| 2e−4 | 56.4 ± 3.2 | 16.5 ± 0.8 | 67.9 ± 1.5 | 20.0 ± 3.0 | 66.7 ± 2.0 | 36.7 ± 1.7 | 44.1 ± 0.8 |
| 5e−4 | 76.3 ± 3.1 | 20.4 ± 0.8 | 74.2 ± 1.2 | 20.8 ± 1.7 | 75.6 ± 0.9 | 41.0 ± 1.8 | 51.4 ± 1.0 |
| 1e−3 | 86.7 ± 2.6 | **23.4 ± 0.6** | 82.2 ± 3.0 | 20.6 ± 1.7 | 81.2 ± 1.1 | 44.9 ± 2.7 | 56.5 ± 0.7 |
| 2e−3 | **87.1 ± 3.2** | 23.2 ± 1.8 | **83.8 ± 2.0** | **21.1 ± 1.5** | **82.2 ± 1.1** | 46.8 ± 2.4 | **57.4 ± 0.7** |
| **Best LR** | 2e−3 | 1e−3 | 2e−3 | 2e−3 | 2e−3 | 2e−3 | 2e−3 |

Table 11: Performance of SIFT_random with `lora_r` = 8 across different learning rates. Results are accuracy (%) ± standard deviation.

| Learning Rate | MultiArith | GSM8K | AddSub | AQuA | SingleEq | SVAMP | Average |
|---|---|---|---|---|---|---|---|
| 2e−5 | – | – | – | – | – | – | – |
| 5e−5 | – | – | – | – | – | – | – |
| 1e−4 | 58.3 ± 3.8 | 16.7 ± 1.4 | 66.3 ± 0.9 | 20.1 ± 2.0 | 69.1 ± 0.9 | 36.5 ± 2.7 | 44.5 ± 0.5 |
| 2e−4 | 73.5 ± 1.5 | 19.2 ± 1.2 | 70.7 ± 0.4 | **21.8 ± 0.8** | 75.7 ± 0.9 | 41.5 ± 1.6 | 50.4 ± 0.1 |
| 5e−4 | **87.8 ± 1.8** | **23.9 ± 1.0** | 82.9 ± 1.4 | 21.7 ± 1.9 | 83.1 ± 1.2 | 45.1 ± 0.8 | **57.4 ± 0.3** |
| 1e−3 | 85.5 ± 2.3 | 23.5 ± 2.1 | **84.0 ± 4.3** | 18.4 ± 1.6 | **83.5 ± 1.9** | **48.1 ± 2.3** | 57.2 ± 1.0 |
| 2e−3 | – | – | – | – | – | – | – |
| **Best LR** | 5e−4 | 5e−4 | 1e−3 | 2e−4 | 1e−3 | 1e−3 | 5e−4 |

Table 12: Performance of SIFT_random with `lora_r` = 16 across different learning rates. Results are accuracy (%) ± standard deviation.

| Learning Rate | MultiArith | GSM8K | AddSub | AQuA | SingleEq | SVAMP | Average |
|---|---|---|---|---|---|---|---|
| 2e−5 | – | – | – | – | – | – | – |
| 5e−5 | – | – | – | – | – | – | – |
| 1e−4 | 74.8 ± 2.0 | 21.1 ± 1.2 | 72.0 ± 1.1 | **23.4 ± 1.9** | 75.9 ± 1.0 | 42.5 ± 0.8 | 51.6 ± 0.2 |
| 2e−4 | 85.2 ± 0.8 | 23.9 ± 1.3 | 81.4 ± 3.2 | 22.8 ± 2.0 | 82.2 ± 0.7 | 45.4 ± 1.9 | 56.8 ± 0.8 |
| 5e−4 | 87.1 ± 4.1 | 25.4 ± 0.7 | 84.6 ± 1.6 | 22.0 ± 1.6 | 84.6 ± 0.9 | 48.7 ± 3.3 | 58.8 ± 0.7 |
| 1e−3 | 92.0 ± 1.7 | 26.1 ± 1.9 | 86.0 ± 1.7 | 22.3 ± 1.6 | 85.6 ± 1.4 | 46.2 ± 2.9 | 59.7 ± 0.8 |
| 2e−3 | – | – | – | – | – | – | – |
| **Best LR** | 1e−3 | 1e−3 | 1e−3 | 1e−4 | 1e−3 | 5e−4 | 1e−3 |

## D.4 SUPER

Table 13: Performance of Super with `lora_r = 4` across different learning rates. Results are accuracy (%) ± standard deviation.

| Learning Rate | MultiArith | GSM8K | AddSub | AQuA | SingleEq | SVAMP | Average |
|---|---|---|---|---|---|---|---|
| 2e−5 | – | – | – | – | – | – | – |
| 5e−5 | – | – | – | – | – | – | – |
| 1e−4 | – | – | – | – | – | – | – |
| 2e−4 | $70.1 \pm 2.7$ | $20.2 \pm 0.8$ | $71.2 \pm 1.1$ | $23.2 \pm 1.6$ | $74.8 \pm 0.9$ | $41.7 \pm 0.9$ | $50.2 \pm 0.9$ |
| 5e−4 | $84.1 \pm 2.9$ | $22.2 \pm 0.4$ | $82.7 \pm 0.7$ | $22.4 \pm 1.3$ | $82.8 \pm 1.0$ | $43.7 \pm 1.7$ | $56.3 \pm 0.3$ |
| 1e−3 | $88.0 \pm 1.9$ | $\mathbf{23.5 \pm 1.5}$ | $85.2 \pm 1.4$ | $23.9 \pm 2.1$ | $\mathbf{84.3 \pm 1.2}$ | $44.3 \pm 1.2$ | $\underline{\mathbf{58.2 \pm 0.5}}$ |
| 2e−3 | $\mathbf{88.9 \pm 0.9}$ | $22.3 \pm 0.4$ | $\underline{\mathbf{85.5 \pm 2.0}}$ | $\mathbf{24.3 \pm 1.8}$ | $82.7 \pm 0.2$ | $\mathbf{44.5 \pm 1.8}$ | $58.0 \pm 0.4$ |
| **Best LR** | 2e−3 | 1e−3 | 2e−3 | 2e−3 | 1e−3 | 2e−3 | 1e−3 |

Table 14: Performance of Super with `lora_r = 8` across different learning rates. Results are accuracy (%) ± standard deviation.

| Learning Rate | MultiArith | GSM8K | AddSub | AQuA | SingleEq | SVAMP | Average |
|---|---|---|---|---|---|---|---|
| 2e−5 | $38.9 \pm 0.7$ | $14.4 \pm 0.4$ | $63.0 \pm 2.4$ | $21.0 \pm 1.5$ | $66.7 \pm 0.3$ | $36.0 \pm 1.5$ | $40.0 \pm 0.8$ |
| 5e−5 | $57.2 \pm 6.4$ | $17.9 \pm 0.2$ | $67.4 \pm 1.5$ | $22.7 \pm 0.5$ | $67.1 \pm 0.4$ | $37.3 \pm 0.5$ | $44.9 \pm 1.0$ |
| 1e−4 | $69.2 \pm 4.4$ | $20.4 \pm 0.6$ | $70.2 \pm 1.7$ | $22.2 \pm 1.4$ | $74.5 \pm 0.3$ | $40.2 \pm 1.5$ | $49.5 \pm 1.2$ |
| 2e−4 | $82.2 \pm 2.0$ | $21.6 \pm 0.8$ | $77.7 \pm 0.5$ | $22.4 \pm 0.4$ | $80.9 \pm 0.8$ | $43.0 \pm 1.8$ | $54.6 \pm 0.5$ |
| 5e−4 | $90.9 \pm 1.5$ | $\mathbf{25.4 \pm 0.9}$ | $85.1 \pm 0.8$ | $\mathbf{22.9 \pm 1.0}$ | $84.4 \pm 1.3$ | $\mathbf{46.7 \pm 0.9}$ | $\mathbf{59.2 \pm 0.3}$ |
| 1e−3 | $\mathbf{91.6 \pm 1.1}$ | $24.2 \pm 2.8$ | $\mathbf{86.3 \pm 3.2}$ | $22.0 \pm 2.0$ | $\mathbf{84.6 \pm 1.7}$ | $44.0 \pm 2.1$ | $58.8 \pm 2.1$ |
| 2e−3 | – | – | – | – | – | – | – |
| **Best LR** | 1e−3 | 5e−4 | 1e−3 | 5e−4 | 1e−3 | 5e−4 | 5e−4 |

Table 15: Performance of Super with `lora_r = 16` across different learning rates. Results are accuracy (%) ± standard deviation.

| Learning Rate | MultiArith | GSM8K | AddSub | AQuA | SingleEq | SVAMP | Average |
|---|---|---|---|---|---|---|---|
| 2e−5 | $49.2 \pm 1.7$ | $16.8 \pm 0.9$ | $64.9 \pm 1.2$ | $22.7 \pm 1.0$ | $67.9 \pm 0.5$ | $37.6 \pm 0.3$ | $43.2 \pm 0.4$ |
| 5e−5 | $66.3 \pm 4.2$ | $20.3 \pm 0.3$ | $70.0 \pm 1.1$ | $23.5 \pm 2.5$ | $72.7 \pm 0.7$ | $40.3 \pm 1.6$ | $48.8 \pm 1.4$ |
| 1e−4 | $83.1 \pm 1.7$ | $22.5 \pm 0.3$ | $78.2 \pm 0.7$ | $22.8 \pm 0.8$ | $79.1 \pm 0.4$ | $43.9 \pm 0.9$ | $54.9 \pm 0.5$ |
| 2e−4 | $87.6 \pm 2.0$ | $\mathbf{24.7 \pm 0.8}$ | $85.2 \pm 0.9$ | $\mathbf{23.7 \pm 3.7}$ | $\mathbf{85.0 \pm 0.4}$ | $\mathbf{47.1 \pm 1.0}$ | $\mathbf{58.9 \pm 0.5}$ |
| 5e−4 | $\mathbf{88.4 \pm 3.8}$ | $24.1 \pm 0.6$ | $\mathbf{85.7 \pm 1.7}$ | $23.4 \pm 0.6$ | $83.7 \pm 0.9$ | $44.1 \pm 2.1$ | $58.2 \pm 0.8$ |
| 1e−3 | $88.3 \pm 5.8$ | $23.6 \pm 1.6$ | $84.6 \pm 1.9$ | $21.5 \pm 1.9$ | $82.7 \pm 0.9$ | $44.3 \pm 0.2$ | $57.5 \pm 0.7$ |
| 2e−3 | – | – | – | – | – | – | – |
| **Best LR** | 5e−4 | 2e−4 | 5e−4 | 2e−4 | 2e−4 | 2e−4 | 2e−4 |

## D.5 SUPER (MATH CALIBRATED)

Table 16: Performance of Super_math_calibr with `lora_r = 4` across different learning rates. Results are accuracy (%) ± standard deviation.

| Learning Rate | MultiArith | GSM8K | AddSub | AQuA | SingleEq | SVAMP | Average |
|---|---|---|---|---|---|---|---|
| 2e−5 | – | – | – | – | – | – | – |
| 5e−5 | – | – | – | – | – | – | – |
| 1e−4 | – | – | – | – | – | – | – |
| 2e−4 | $72.2 \pm 2.0$ | $20.5 \pm 1.0$ | $71.6 \pm 1.8$ | $\mathbf{23.3 \pm 1.2}$ | $76.3 \pm 1.7$ | $41.3 \pm 1.3$ | $50.9 \pm 0.7$ |
| 5e−4 | $82.4 \pm 2.5$ | $23.1 \pm 0.5$ | $80.8 \pm 1.8$ | $23.1 \pm 1.5$ | $82.4 \pm 1.4$ | $\mathbf{45.4 \pm 0.9}$ | $56.2 \pm 0.6$ |
| 1e−3 | $86.8 \pm 3.8$ | $\underline{\mathbf{24.9 \pm 1.4}}$ | $82.4 \pm 1.2$ | $23.0 \pm 2.7$ | $82.1 \pm 0.9$ | $42.8 \pm 1.5$ | $57.0 \pm 0.9$ |
| 2e−3 | $\mathbf{91.2 \pm 1.3}$ | $24.2 \pm 0.7$ | $\mathbf{83.2 \pm 2.6}$ | $22.3 \pm 1.9$ | $\mathbf{83.1 \pm 1.8}$ | $41.4 \pm 2.8$ | $\mathbf{57.6 \pm 1.0}$ |
| **Best LR** | 2e−3 | 1e−3 | 2e−3 | 2e−4 | 2e−3 | 5e−4 | 2e−3 |

Table 17: Performance of Super_math_calibr with `lora_r = 8` across different learning rates. Results are accuracy (%) ± standard deviation.

| Learning Rate | MultiArith | GSM8K | AddSub | AQuA | SingleEq | SVAMP | Average |
|---|---|---|---|---|---|---|---|
| 2e−5 | – | – | – | – | – | – | – |
| 5e−5 | – | – | – | – | – | – | – |
| 1e−4 | $70.1 \pm 2.6$ | $21.4 \pm 0.1$ | $71.7 \pm 1.4$ | $22.7 \pm 0.6$ | $75.5 \pm 1.3$ | $40.6 \pm 1.1$ | $50.3 \pm 0.8$ |
| 2e−4 | $79.1 \pm 1.1$ | $22.4 \pm 0.2$ | $77.6 \pm 2.4$ | $\mathbf{24.0 \pm 1.6}$ | $81.6 \pm 0.6$ | $44.4 \pm 0.3$ | $54.8 \pm 0.7$ |
| 5e−4 | $88.1 \pm 1.5$ | $\mathbf{25.2 \pm 0.9}$ | $\mathbf{84.5 \pm 1.9}$ | $23.3 \pm 1.4$ | $\mathbf{85.9 \pm 0.8}$ | $\mathbf{46.3 \pm 1.0}$ | $\mathbf{58.7 \pm 0.6}$ |
| 1e−3 | $\mathbf{88.8 \pm 2.7}$ | $23.7 \pm 0.8$ | $83.1 \pm 1.6$ | $21.8 \pm 0.8$ | $82.7 \pm 1.9$ | $43.5 \pm 3.3$ | $57.3 \pm 0.8$ |
| 2e−3 | – | – | – | – | – | – | – |
| **Best LR** | 1e−3 | 5e−4 | 5e−4 | 2e−4 | 5e−4 | 5e−4 | 5e−4 |

Table 18: Performance of Super_math_calibr with `lora_r = 16` across different learning rates. Results are accuracy (%) ± standard deviation.

| Learning Rate | MultiArith | GSM8K | AddSub | AQuA | SingleEq | SVAMP | Average |
|---|---|---|---|---|---|---|---|
| 2e−5 | – | – | – | – | – | – | – |
| 5e−5 | – | – | – | – | – | – | – |
| 1e−4 | $81.9 \pm 2.9$ | $23.2 \pm 0.2$ | $76.6 \pm 1.9$ | $22.2 \pm 0.9$ | $80.5 \pm 0.8$ | $43.8 \pm 2.0$ | $54.7 \pm 1.0$ |
| 2e−4 | $88.1 \pm 2.1$ | $\mathbf{25.0 \pm 0.7}$ | $83.8 \pm 1.9$ | $\mathbf{23.5 \pm 1.9}$ | $\mathbf{85.4 \pm 0.8}$ | $\mathbf{47.8 \pm 1.2}$ | $\mathbf{59.0 \pm 1.1}$ |
| 5e−4 | $89.6 \pm 2.5$ | $24.4 \pm 0.8$ | $82.8 \pm 4.3$ | $21.4 \pm 3.0$ | $83.9 \pm 1.8$ | $44.9 \pm 4.0$ | $57.8 \pm 1.8$ |
| 1e−3 | $\mathbf{92.7 \pm 0.1}$ | $22.5 \pm 1.4$ | $\mathbf{84.3 \pm 2.8}$ | $21.9 \pm 0.6$ | $82.5 \pm 2.8$ | $42.1 \pm 2.3$ | $57.7 \pm 1.5$ |
| 2e−3 | – | – | – | – | – | – | – |
| **Best LR** | 1e−3 | 2e−4 | 1e−3 | 2e−4 | 2e−4 | 2e−4 | 2e−4 |

## D.6 SUPRA (0.3)

Table 19: Performance of Supra_0.3 with `lora_r = 4` across different learning rates. Results are accuracy (%) ± standard deviation.

| Learning Rate | MultiArith | GSM8K | AddSub | AQuA | SingleEq | SVAMP | Average |
|---|---|---|---|---|---|---|---|
| 2e−5 | – | – | – | – | – | – | – |
| 5e−5 | – | – | – | – | – | – | – |
| 1e−4 | – | – | – | – | – | – | – |
| 2e−4 | – | – | – | – | – | – | – |
| 5e−4 | 92.1 ± 2.0 | 23.2 ± 1.2 | 81.4 ± 1.1 | 23.4 ± 2.9 | 82.2 ± 1.7 | **42.7 ± 1.6** | 57.5 ± 1.0 |
| 1e−3 | **92.5 ± 3.0** | **24.8 ± 1.1** | 82.4 ± 1.8 | **23.6 ± 1.0** | **82.7 ± 1.1** | 42.2 ± 3.2 | **58.0 ± 0.7** |
| 2e−3 | 90.4 ± 2.5 | 22.9 ± 1.2 | **82.4 ± 2.1** | 21.7 ± 1.4 | 80.8 ± 2.4 | 41.9 ± 2.6 | 56.7 ± 0.7 |
| **Best LR** | 1e−3 | 1e−3 | 2e−3 | 1e−3 | 1e−3 | 5e−4 | 1e−3 |

Table 20: Performance of Supra_0.3 with `lora_r = 8` across different learning rates. Results are accuracy (%) ± standard deviation.

| Learning Rate | MultiArith | GSM8K | AddSub | AQuA | SingleEq | SVAMP | Average |
|---|---|---|---|---|---|---|---|
| 2e−5 | – | – | – | – | – | – | – |
| 5e−5 | – | – | – | – | – | – | – |
| 1e−4 | – | – | – | – | – | – | – |
| 2e−4 | – | – | – | – | – | – | – |
| 5e−4 | 91.8 ± 1.1 | **25.8 ± 0.7** | 86.4 ± 0.6 | **25.1 ± 1.8** | **85.7 ± 0.7** | **47.5 ± 1.4** | **60.4 ± 0.6** |
| 1e−3 | **93.6 ± 0.5** | 24.9 ± 1.2 | **87.0 ± 0.5** | 23.9 ± 1.1 | 85.6 ± 2.2 | 46.9 ± 1.2 | 60.3 ± 0.5 |
| 2e−3 | 92.1 ± 0.5 | 21.9 ± 0.6 | 82.0 ± 4.6 | 23.1 ± 0.8 | 81.3 ± 2.4 | 39.1 ± 3.0 | 56.6 ± 1.5 |
| **Best LR** | 1e−3 | 5e−4 | 1e−3 | 5e−4 | 5e−4 | 5e−4 | 5e−4 |

## D.7 SUPRA (0.3) (MATH CALIBRATED)

Table 21: Performance of Supra_0.3_math_calibr with `lora_r = 4` across different learning rates. Results are accuracy (%) where only one learning rate was tested.

| Learning Rate | MultiArith | GSM8K | AddSub | AQuA | SingleEq | SVAMP | Average |
|---|---|---|---|---|---|---|---|
| 2e−5 | – | – | – | – | – | – | – |
| 5e−5 | – | – | – | – | – | – | – |
| 1e−4 | – | – | – | – | – | – | – |
| 2e−4 | – | – | – | – | – | – | – |
| 5e−4 | **86.8** | **24.1** | **82.3** | **20.9** | **83.7** | **44.4** | **57.0** |
| 1e−3 | – | – | – | – | – | – | – |
| 2e−3 | – | – | – | – | – | – | – |
| **Best LR** | 5e−4 | 5e−4 | 5e−4 | 5e−4 | 5e−4 | 5e−4 | 5e−4 |

## D.8 SUPRA (0.55)

Table 22: Performance of Supra_0.55 with `lora_r` = 4 across different learning rates. Results are accuracy (%) ± standard deviation.

| Learning Rate | MultiArith | GSM8K | AddSub | AQuA | SingleEq | SVAMP | Average |
|---|---|---|---|---|---|---|---|
| 2e−5 | – | – | – | – | – | – | – |
| 5e−5 | – | – | – | – | – | – | – |
| 1e−4 | – | – | – | – | – | – | – |
| 2e−4 | – | – | – | – | – | – | – |
| 5e−4 | 90.9 ± 1.4 | 22.3 ± 1.2 | 81.9 ± 0.8 | 21.8 ± 0.9 | 81.5 ± 0.7 | **43.8 ± 1.3** | 57.1 ± 0.2 |
| 1e−3 | **93.1 ± 1.8** | **24.1 ± 0.5** | 82.4 ± 4.0 | **22.7 ± 2.8** | **83.5 ± 0.9** | 41.3 ± 1.3 | **57.8 ± 1.1** |
| 2e−3 | 92.9 ± 1.8 | 22.7 ± 0.5 | **82.9 ± 2.8** | 22.2 ± 3.9 | 82.2 ± 0.6 | 43.1 ± 1.0 | 57.7 ± 0.8 |
| **Best LR** | 1e−3 | 1e−3 | 2e−3 | 1e−3 | 1e−3 | 5e−4 | 1e−3 |

Table 23: Performance of Supra_0.55 with `lora_r` = 8 across different learning rates. Results are accuracy (%) ± standard deviation.

| Learning Rate | MultiArith | GSM8K | AddSub | AQuA | SingleEq | SVAMP | Average |
|---|---|---|---|---|---|---|---|
| 2e−5 | – | – | – | – | – | – | – |
| 5e−5 | – | – | – | – | – | – | – |
| 1e−4 | – | – | – | – | – | – | – |
| 2e−4 | – | – | – | – | – | – | – |
| 5e−4 | **94.8 ± 1.8** | **25.9 ± 1.4** | 85.6 ± 2.5 | **23.8 ± 0.9** | **86.5 ± 0.7** | **47.1 ± 3.6** | **60.6 ± 1.2** |
| 1e−3 | 93.0 ± 2.3 | 24.9 ± 0.7 | **87.1 ± 1.2** | 22.2 ± 1.6 | 84.9 ± 0.5 | 43.2 ± 3.8 | 59.2 ± 0.4 |
| 2e−3 | 92.5 ± 0.9 | 21.2 ± 0.9 | 84.2 ± 1.4 | 23.2 ± 1.8 | 82.1 ± 2.6 | 42.3 ± 1.7 | 57.6 ± 0.8 |
| **Best LR** | 5e−4 | 5e−4 | 1e−3 | 5e−4 | 5e−4 | 5e−4 | 5e−4 |

## D.9 SUPRA (0.8)

Table 24: Performance of Supra_0.8 with `lora_r` = 4 across different learning rates. Results are accuracy (%) ± standard deviation.

| Learning Rate | MultiArith | GSM8K | AddSub | AQuA | SingleEq | SVAMP | Average |
|---|---|---|---|---|---|---|---|
| 2e−5 | – | – | – | – | – | – | – |
| 5e−5 | – | – | – | – | – | – | – |
| 1e−4 | – | – | – | – | – | – | – |
| 2e−4 | – | – | – | – | – | – | – |
| 5e−4 | 90.7 ± 1.4 | **22.2 ± 1.8** | 81.4 ± 1.4 | 23.1 ± 0.6 | **82.5 ± 1.4** | 43.1 ± 1.6 | **57.2 ± 0.6** |
| 1e−3 | **91.7 ± 0.7** | 21.6 ± 1.6 | 81.5 ± 3.3 | **24.8 ± 2.8** | 81.1 ± 5.3 | 41.7 ± 4.7 | 57.1 ± 2.2 |
| 2e−3 | 90.2 ± 1.9 | 21.6 ± 1.1 | **81.6 ± 3.0** | 20.7 ± 3.0 | 82.3 ± 1.4 | **43.2 ± 1.1** | 56.6 ± 0.8 |
| **Best LR** | 1e−3 | 5e−4 | 2e−3 | 1e−3 | 5e−4 | 2e−3 | 5e−4 |

Table 25: Performance of Supra_0.8 with `lora_r` $= 8$ across different learning rates. Results are accuracy (%) $\pm$ standard deviation.

| Learning Rate | MultiArith | GSM8K | AddSub | AQuA | SingleEq | SVAMP | Average |
|---|---|---|---|---|---|---|---|
| 2e$-$5 | – | – | – | – | – | – | – |
| 5e$-$5 | – | – | – | – | – | – | – |
| 1e$-$4 | – | – | – | – | – | – | – |
| 2e$-$4 | – | – | – | – | – | – | – |
| 5e$-$4 | **95.1 $\pm$ 1.5** | **26.0 $\pm$ 0.6** | 85.6 $\pm$ 2.7 | **22.7 $\pm$ 1.1** | 84.8 $\pm$ 1.1 | 45.5 $\pm$ 2.3 | 59.9 $\pm$ 0.9 |
| 1e$-$3 | 93.2 $\pm$ 0.8 | 25.1 $\pm$ 0.7 | **88.5 $\pm$ 1.0** | 21.4 $\pm$ 1.8 | **87.2 $\pm$ 0.9** | **48.3 $\pm$ 1.5** | **60.6 $\pm$ 0.3** |
| 2e$-$3 | 92.5 $\pm$ 1.8 | 21.4 $\pm$ 1.2 | 82.2 $\pm$ 2.2 | 20.2 $\pm$ 4.1 | 82.5 $\pm$ 1.3 | 42.0 $\pm$ 2.5 | 56.8 $\pm$ 1.5 |
| **Best LR** | 5e$-$4 | 5e$-$4 | 1e$-$3 | 5e$-$4 | 1e$-$3 | 1e$-$3 | 1e$-$3 |

## D.10 HIGHER RANK: $r = 16$

Table 26: Comparison of methods with `lora_r` $= 16$. Results are accuracy (%) $\pm$ standard deviation.

| Method | MultiArith | GSM8K | AddSub | AQuA | SingleEq | SVAMP | Average |
|---|---|---|---|---|---|---|---|
| LoRA | 93.3 $\pm$ 1.6 | 24.8 $\pm$ 1.8 | 84.7 $\pm$ 5.5 | 23.5 $\pm$ 2.2 | 82.1 $\pm$ 5.7 | 45.8 $\pm$ 4.0 | 58.1 $\pm$ 4.0 |
| SIFT | **93.5 $\pm$ 1.4** | 24.0 $\pm$ 0.8 | **87.0 $\pm$ 0.7** | **24.3 $\pm$ 0.8** | 84.1 $\pm$ 1.2 | 44.8 $\pm$ 1.6 | 59.0 $\pm$ 1.1 |
| SUPER | 88.4 $\pm$ 3.8 | 24.7 $\pm$ 0.8 | 85.7 $\pm$ 1.7 | 23.7 $\pm$ 3.7 | 85.0 $\pm$ 0.4 | 47.1 $\pm$ 1.0 | 58.9 $\pm$ 0.5 |
| SIFT_Random | 92.0 $\pm$ 1.7 | **26.1 $\pm$ 1.9** | 86.0 $\pm$ 1.7 | 23.4 $\pm$ 1.9 | **85.6 $\pm$ 1.4** | **48.7 $\pm$ 3.3** | **59.7 $\pm$ 0.8** |
| SUPER_Math_Calibr | 92.7 $\pm$ 0.1 | 25.0 $\pm$ 0.7 | 84.3 $\pm$ 2.8 | 23.5 $\pm$ 1.9 | 85.4 $\pm$ 0.8 | 47.8 $\pm$ 1.2 | 59.0 $\pm$ 1.1 |
| Supra_0.3 | – | – | – | – | – | – | – |
| Supra_0.3_Math_Calibr | – | – | – | – | – | – | – |
| Supra_0.55 | – | – | – | – | – | – | – |
| Supra_0.8 | – | – | – | – | – | – | – |

