# OpenReview forum: "Supra-Tuning: Combining Outlier and Low-Rank Adaptation for Sparse and Efficient LLM Fine-Tuning"
_ICLR.cc/2026/Conference — Submitted to ICLR 2026_

### Official Review · Reviewer_vVKC · 2025-10-26

**Soundness:** 2
**Presentation:** 2
**Contribution:** 1
**Rating:** 2
**Confidence:** 3

**Summary:**

The paper presents Super, which selects and trains only a small set of influential weights, and proposes Supra, which combines with LoRA.
They demonstrate strong performance in downstream tasks while reducing computational and memory overhead.

**Strengths:**

- Super uses the WANDA saliency computed in a single forward pass, which is easy to reproduce and deploy at scale.

**Weaknesses:**

- limited novelty: Both outlier metrics (e.g., WANDA) and LoRA are known; the main idea is to combine them.
- Narrow evaluation scope: Experiments are restricted to the math dataset.

**Questions:**

- r = 8 and r = 4 are described as “moderate” and “low” budgets, but could the authors provide actual metrics (e.g., training time, GPU memory, or FLOPs) to quantify the real difference in efficiency?
- How sensitive is Super/Supra to the WANDA ranking?
- Could you include computational cost study (e.g., wall-clock time) for Super, LoRA, and Supra to substantiate claims of reduced overhead?
- Since evaluation is math-centric, how do results transfer to other tasks?
- Lack of study on equation 7.

---

### Official Review · Reviewer_pxQb · 2025-10-29

**Soundness:** 2
**Presentation:** 2
**Contribution:** 2
**Rating:** 2
**Confidence:** 5

**Summary:**

The paper introduces Supra, a parameter-efficient tuning method with two parts: **Super** selects a small set of high-impact “super weights” using a WANDA-style outlier score and trains only those, while **Supra** unifies Super with LoRA by allocating a fixed per-layer parameter budget between sparse and low-rank updates through a simple ratio α. Experiments show that Super excels under tight budgets and Supra dominates when more parameters are available, outperforming existing baselines.

**Strengths:**

1. The method is conceptually simple and enables fast division of trainable parameters into sparse and low-rank parts.
2. The paper provides detailed ablation studies on hyperparameters and reports variance statistics, enhancing the credibility of the results.

**Weaknesses:**

1. The parameter selection strategy follows the same principle as WANDA, offering limited novelty.
2. The comparative analysis is somewhat narrow, as the paper only benchmarks against basic ablation variants such as LoRA, SIFT, and RoSA. Comparisons with more recent methods (e.g., NeFT, SpIEL, DoRA, HiRA) would strengthen the claims.
3. The evaluation datasets are relatively limited, and performance reported on GSM8K is less than similar works.
4. Model configurations are not clear. The details on LLaMA-3 (1B/3B/8B) settings are missing.
5. Several tables have formatting issues (lines extending beyond page margins).
6. Appendix C appears incomplete.

**Questions:**

Beyond the issues mentioned above, a key question concerns the sample selection for the WANDA metric. How were these samples chosen, and does the approach generalize well to other tasks and domains?

---

### Official Review · Reviewer_mwmk · 2025-11-01

**Soundness:** 2
**Presentation:** 2
**Contribution:** 2
**Rating:** 2
**Confidence:** 3

**Summary:**

This paper proposes two efficient LLM fine-tuning methods: Super and Supra. Super uses the WANDA metric to filter outlier weights for sparse fine-tuning, while Supra combines Super with LoRA to achieve a fusion of sparsity and low-rank adaptation. Experiments show that both methods require fewer parameters and outperform existing methods on mathematical reasoning tasks.

**Strengths:**

- Two efficient algorithms, Super and Supra, are proposed, eliminating the dependency on LoRA's specific parameter structure updates.

- Figure 1 clearly explains the algorithms proposed by the authors.

**Weaknesses:**

- Appendix C, “ADDITIONAL EXPERIMENTS,” contains only a title and no content.

- Appendix D's tables are missing numerous experimental results without any explanation.

- Table 2's “GSM8K” column does not highlight the best results.

- Only the main experiment is shown; further analytical experiments are missing.

**Questions:**

- Compared to the baseline method, how efficient are Super and Supra? Aren't they faster?

- When training with Super and Supra, does the effective rank of each matrix in the model change during training?

---

### Official Review · Reviewer_HyVw · 2025-11-03

**Soundness:** 2
**Presentation:** 2
**Contribution:** 2
**Rating:** 2
**Confidence:** 5

**Summary:**

The paper presents Super, a sparse fine-tuning method selecting outlier weights via the WANDA metric, and Supra, a hybrid combining Super with LoRA for large language model adaptation. Experiments on the Math10K dataset and six math reasoning benchmarks show modest gains over existing PEFT baselines such as LoRA and SIFT.

**Strengths:**

- The motivation of combining sparse and low-rank adaptation is reasonable and aligns with emerging trends in efficient LLM fine-tuning.

**Weaknesses:**

- Writing is poor. The text is verbose, repetitive, and lacks clarity in technical exposition. Many notations, equations, and definitions are redundant or inconsistently introduced. For example, Sec 1.1 said that all notations will be present in Section A, but Sec 3.1 introduces notations. Algorithm 1 only has two lines, which is strange.

The writing showed this work is away from completeness with proper proof-read.

**Questions:**

See the comment.

---

### Meta-Review · Area_Chair_udEx · 2025-12-28

**Summary:**

This paper proposes Super, a sparse PEFT method based on WANDA-style outlier weight selection, and Supra, a hybrid approach that combines sparse updates with LoRA under a fixed per-layer parameter budget. Across reviews, there is agreement that the empirical trend is sensible—sparse updates perform best under tight budgets, while hybrid sparse + low-rank adaptation performs better when more parameters are available. However, reviewers consistently raised concerns about limited novelty relative to existing sparse and hybrid PEFT methods, narrow evaluation scope (math-only benchmarks), missing or incomplete experimental sections, and insufficient reporting of computational efficiency metrics. Presentation quality and clarity were also flagged by multiple reviewers. While some reviewer comments were imprecise or based on misunderstandings of standard PEFT mechanisms, the core concerns regarding novelty, experimental completeness, and generality remain substantial and informed the final recommendation.

**Reviewer Concerns:**

Novelty: The core methods rely directly on existing components (WANDA saliency and LoRA), and the rebuttal did not sufficiently establish a clear conceptual or theoretical advance over prior hybrid PEFT approaches (e.g., RoSA, SLTrain).

Evaluation scope: Experiments remain limited to Math10K and arithmetic reasoning tasks, leaving generalization to other domains untested.

Efficiency claims: No wall-clock time, memory usage, or FLOPs analysis was added to substantiate claims of improved efficiency.

Presentation quality: While some explanations improved, the paper still suffers from redundancy, organizational issues, and formatting problems.

**Reviewer Scores:**

Reviewer HyVw: Likely unchanged
Reviewer mwmk: Likely unchanged
Reviewer pxQb: Likely unchanged
Reviewer vVKC: Likely unchanged

---

### Decision · Program_Chairs · 2026-01-26

Reject